# Impact of Alternatively Polyadenylated Isoforms of ETHYLENE RESPONSE FACTOR4 with Activator and Repressor Function on Senescence in *Arabidopsis thaliana* L.

**DOI:** 10.3390/genes10020091

**Published:** 2019-01-28

**Authors:** Lena Riester, Siliya Köster-Hofmann, Jasmin Doll, Kenneth W. Berendzen, Ulrike Zentgraf

**Affiliations:** Center for Plant Molecular Biology (ZMBP), University of Tuebingen, 72076 Tuebingen, Germany; lena.riester@zmbp.uni-tuebingen.de (L.R.); hofmann@uni-tuebingen.de (S.K.-H.); jasmin.doll@zmbp.uni-tuebingen.de (J.D.); kenneth.berendzen@zmbp.uni-tuebingen.de (K.W.B.)

**Keywords:** Alternative polyadenylation, ETHYLENE RESPONSE FACTOR4, senescence regulation, CATALASE3, FPA, Arabidopsis

## Abstract

Leaf senescence is highly regulated by transcriptional reprogramming, implying an important role for transcriptional regulators. ETHYLENE RESPONSE FACTOR4 (ERF4) was shown to be involved in senescence regulation and to exist in two different isoforms due to alternative polyadenylation of its pre-mRNA. One of these isoforms, ERF4-R, contains an ERF-associated amphiphilic repression (EAR) motif and acts as repressor, whereas the other form, ERF4-A, is lacking this motif and acts as activator. Here, we analyzed the impact of these isoforms on senescence. Both isoforms were able to complement the delayed senescence phenotype of the *erf4* mutant with a tendency of ERF4-A for a slightly better complementation. However, overexpression led to accelerated senescence of 35S:*ERF4-R* plants but not of 35S:*ERF4-A* plants. We identified *CATALASE3* (*CAT3*) as direct target gene of ERF4 in a yeast-one-hybrid screen. Both isoforms directly bind to the *CAT3* promoter but have antagonistic effects on gene expression. The ratio of *ERF4-A* to *ERF4-R* mRNA changed during development, leading to a complex age-dependent regulation of CAT3 activity. The RNA-binding protein FPA shifted the R/A-ratio and *fpa* mutants are pointing towards a role of alternative polyadenylation regulators in senescence.

## 1. Introduction

At the end of their development, plants lose their photosynthetic capacity and finally shed their leaves. To avoid an uneconomic loss of stored energy, minerals and nutrients through leaf abscission, the leaves undergo leaf senescence before they die. During this process, plants recycle resources and nutrients by relocating them to developing organs. It is a dynamic and highly regulated process driven by a complex genetically encoded program. However, under stress conditions, senescence is induced prematurely as an exit strategy. Therefore, this program requires a high plasticity through constantly integrating endogenous and exogenous signals. Molecules such as plant hormones, nitrogen and sugar compounds, calcium, reactive oxygen species (ROS) and most likely further substances mediate signal transduction [1,2,3,4,5].

In general, ROS act as signaling molecules at their site of production due to their short half-life. Among all ROS, hydrogen peroxide (H_2_O_2_) is most likely the signaling molecule with the broadest reach, since it is more stable than other ROS and it can pass membranes [6]. Hydrogen peroxide is involved in many signal transduction pathways spanning from biotic to abiotic stress responses, but also participates in the regulation of senescence processes. The onset of senescence coincides with an increase of intracellular H_2_O_2_ levels as a result of a coordinated regulation of the H_2_O_2_ scavenging enzymes catalase and ascorbate peroxidase [7,8]. Many senescence-associated genes and transcription factors are upregulated by ROS. Differential gene expression plays an important role in the coordination of leaf senescence onset and progression [9,10,11,12]. Breeze and coworkers [12] suggested that the temporal expression and activation of transcription factors (TFs) is a crucial aspect in the course of leaf senescence. Among others, several members of the APETALA2/ETHYLENE RESPONSE FACTOR (AP2/ERFs) family are regulated in a senescence-dependent manner [9,11,12]. AP2/ERFs form a superfamily of 147 TF genes in Arabidopsis [13,14]. Up to now, a number of AP2/ERFs have been reported to be involved in stress responses and developmental processes [15]. Some AP2/ ERF TFs are involved in the responses to components of stress signal transduction pathways, such as ROS, ethylene, jasmonic acid (JA), abscisic acid (ABA) and cytokinin, all of which are also important molecules in senescence-associated signaling [14,16]. During leaf senescence, RELATED TP ABI3/ VP1 (RAV1) and C-REPEAT/ DEHYDRATION RESPONSIBLE ELEMENT BINDING FACTOR 2 (CBF2) have been reported to be positive and negative regulators, respectively. Moreover, Koyama et al. [17] have shown that ERF4 (At3g15210) and ERF8 (At1g53170) act together as positive regulators of leaf senescence by suppressing the expression of its direct target gene *EPITHIOSPECIFIER PROTEIN/ EPITHIOSPECIFYING SENESCENCE REGULATOR* (*ESP/ESR*) (At1g54040). This protein is a negative regulator of the transcription factor WKRY53 (At4g23810), a positive regulator of leaf senescence [18,19]. WRKY53 turns out to be a central node of a complex regulatory network of leaf senescence and to underlie a tight multi-layer control of expression, activity and protein stability [20,21,22,23]. In addition, ERF4 can up-regulate intracellular ROS production, as overexpression of *ERF4* led to higher ROS levels, visualized by Trypan blue and DAB (3,3-diaminobenzidine)-staining [17]. In contrast, *erf4/erf8* double-mutant plants produce fewer amounts of ROS during dark-induced senescence, indicating that ERF4 and/or 8 are themselves involved in regulating intracellular ROS contents [17]. 

The Arabidopsis *ERF4* belongs to the ERF subfamily containing 122 genes. They only possess one AP2/ERF domain, which is their distinguishing feature [13,14]. They can be further classified into 12 subgroups based on further conserved amino acid motifs [14]. *AtERF4* is a member of the group VIIIa ERFs, characterized by the ERF-associated amphiphilic repression (EAR) motif. This motif allows them to act as transcriptional repressors. They can repress target gene transcription even in the presence of ERF activators in transient reporter gene assays [24,25,26]. However, due to alternative polyadenylation and splicing, ERF4 exists in two different protein isoforms, one containing the EAR-motif (ERF4-R), one lacking it (ERF4-A) [27]. Formation of the ERF4-A isoform was shown to be induced by flg22 treatment, a bacterial peptide derived from flagellin, which induces PAMP-triggered immunity and a ROS burst. The plant RNA-binding protein FPA (At2g43410) can inhibit the formation of the ERF4-A isoform and the ROS burst. The ability of *ERF4* to switch from a repressor to an activator by alternative splicing and polyadenylation adds an extra layer of complexity to molecular mechanisms underlying the ERF-mediated gene regulation [27]. 

Alternative splicing (AS) is an important factor in gene regulation. Many transcription factor genes undergo this process, which results in the production of multiple proteins from one single gene [28,29,30,31]. It is involved in a variety of plant growth and developmental processes, such as induction of flowering [32], plant responses to changing environmental conditions and pathogen attacks [28]. However, AS in leaf senescence has so far not been studied in detail. This study aims to analyze the role of alternative splicing and polyadenylation of *ERF4* in leaf senescence. By complementation of the *erf4* mutant plants with both isoforms, we provide evidence that both ERF4 isoforms function in senescence. ERF4-A acts as transcriptional activator and ERF4-R as repressor of their direct target gene *CATALASE3* (*CAT3*), which we could identify in a yeast-one-hybrid (Y1H) screen. Catalase proteins are important enzymes in catalyzing the decomposition of H_2_O_2_ to water (H_2_O) and oxygen (O_2_) and controlling the concentrations of ROS in cells [33,34]. All three Arabidopsis catalases are regulated in a senescence-associated manner [7,8] and H_2_O_2_ is used as signaling molecule in senescence. Moreover, we could show that the ratio of ERF4-A and ERF4-R expression changes over development and is influenced by the RNA-binding protein FPA. A complex pattern of activating and repressing activities on CAT3 function became obvious in the *erf4* mutant plants. We provide a model on how the interplay of the different components might be organized.

## 2. Material and Methods

### 2.1. Yeast-One-Hybrid System

The Matchmaker yeast-one-hybrid library screening system (ClonTech, Heidelberg, Germany) was used to screen for genes that bind to fragments of the *CAT3* promoter. A 150-bp fragment (pos. -182 to -332) upstream of the linear reporter plasmid was integrated into *Saccharomyces cerevisiae* strain Y187 by recombination. The yeast strain with the integrated *pHISi* plasmid was mated with the yeast strain AH109 carrying a cDNA library, which was prepared from RNA of 7-week-old rosette leaves of *Arabidopsis thaliana* and integrated into the *pGADT7-Rec* vector with the Gal4 activation domain and the *leu2* selection marker. The one-hybrid screenings and assays were performed as described in the manufacturer’s protocols. Two independent clones partially coding for *ERF4* clones were found in the screening. Therefore, the full-length cDNA of *ERF4* was cloned into *pGADT7-Rec vector* and transformed again into Y187 cells carrying the *CAT3* driven *HIS3* gene to confirm the interaction.

### 2.2. Expression and Extraction of Recombinant Proteins for DNA-Protein-Interaction- Enzyme-linked Immunosorbent Assay (DPI-ELISA), Pull-Down Assay and in vitro Protein Degradation Assay

The coding sequences of the two ERF4 isoforms [27] were cloned into the expression vector pETG-10A with N-terminal hexahistidine-tag (6xHIS-tag) and in the pDEST-15 expression vector with N-terminal Gluthathione-S-Transferase (GST)-tag (for pull-down assay). For protein expression, the *E. coli* strain BL21 Rosetta was used. The transformed cells were grown in 5 ml selective LB (lysogeny broth) medium overnight and the next day a fresh 50 ml culture was inoculated with 500 μL of the overnight culture and grown in selective LB medium until it reached OD_595_ of 0.5. Protein expression was induced by adding IPTG (Isopropyl-β-D-thiogalactopyranoside) to a final concentration of 1 mM. After 4 h of shaking at 30 °C the cells were harvested by centrifugation (4600 rpm, 20 min, 4 °C). The pellet was resuspended in protein extraction buffer (500 mM NaCl, 5 mM Imidazole, 20 mM Tris-HCl, 2 mM sodium azide, pH 7.0, 1 x complete proteinase inhibitor without ethylenediaminetetraacetic acid (Roche, Basel, Switzerland) was added freshly). Proteins were extracted under native conditions by sonicating and the protein concentration was measured in a Bradford assay (Bio-Rad, Munich, Germany).

### 2.3. DPI-ELISA

Streptavidin-coated, transparent 96-well ELISA plates (Thermo Scientific, Waltham, MA, USA) were used to immobilize double-stranded (ds) oligonucleotides (Biomers, Ulm, Germany). The 5’biotinylated forward and its complementary reverse oligonucleotide were annealed at 95 °C for 5 min, followed by a slow, stepwise cooling down to room temperature. 2 pmol ds oligonucleotides were added per well in a total volume of 60 μL and incubated for 1 h. The liquid was removed after each step. After 1 h of blocking (60 μL Roche blocking solution), 5, 10 and 25 μg of protein crude extracts were diluted with protein dilution buffer (4 mM Hepes pH 7.5, 100 mM KCl, 8% (v/v) glycerol) and the mixture was incubated for 1 h. Two washing steps with 100 μL Qiagen wash buffer were performed, 60 μL of Penta-His-horseradish peroxidase conjugate antibody (Qiagen, Venlo, The Netherlands) was added (diluted 1:1500 in Qiagen wash buffer) and incubated for 1 h. After another two washing steps, the interaction was visualized using the substrate of the peroxidase, o-phenylenediamine dihydrochloride (OPD tablets, Thermo Scientific). This resulted in a yellow product, which was detected in a plate reader (TECAN Safire SFluor4) at 492 nm.

### 2.4. Pull-Down Assay

GST-ERF4-A (49.6 kDa), GST-ERF4-R (51.7 kDa), 6xHIS-ERF4-A (24.5 kDa) and 6xHIS-ERF4-R (26.6 kDa) fusion proteins were expressed as previously described. 500 μg total protein of the crude extracts were incubated at room temperature for 1 h with continuous shaking. HIS-select nickel magnetic agarose beads (Sigma-Aldrich, St. Louis, MO, USA) were used according to manufacturer’s instructions to pull on the 6xHIS-tagged fusion proteins (wash buffer: 50 mM NaH_2_PO_4_, 300 mM NaCl, 10 mM imidazole, elution buffer: 50 mM sodium phosphate pH 8,0, 0,3 M NaCl, 300 mM imidazole). The eluted fraction was analyzed by SDS-PAGE using a monoclonal anti-Glutathione-S-Transferase (GST) antibody (Sigma-Aldrich) and semi-dry Western Blot.

### 2.5. In Vitro Protein Degradation Assay

Leaf 7–8 and 9–13 of 49-day-old Arabidopsis Col-0 plants were harvested separately. Recombinant 6xHIS-ERF4-A and 6xHIS-ERF4-R fusion proteins were produced in bacteria as described earlier. The protein isolation buffer was exchanged by using Amicon Ultra-4 Centrifugal Filter Devices (Merck Millipore, Burlington, MA, USA) with a solution containing 20 mM Tris (pH 8.0) and 100 mM NaCl. The cell-free in vitro degradation assay was modified from Reference [35]. The leaf material was ground in liquid nitrogen and the powder was suspended in a buffer containing 50 mM Tris (pH 7.5), 10 mM NaCl, 10 mM MgCl_2_, 5 mM dithiothreitol and 2 mM ATP. After centrifugation (14,000 rpm, 10 min, 4 °C) 25 μg total protein of the bacterial crude lysate were added into the plant extracts and incubated at room temperature for 0, 15, 30, 60 and 120 min. Subsequently, SDS loading buffer was added and the samples were boiled at 95 °C to terminate the reaction and to denature the proteins. The proteins were detected by SDS-PAGE and immunoblot analysis with a penta-HIS-HRP conjugate antibody (Qiagen). *E. coli* extracts were used as negative control.

### 2.6. Protoplast Leaf Transfection

The transient GUS (ß-glucuronidase) reporter gene assays and BiFC were performed in protoplasts, which were derived from cell suspension culture cells of *Arabidopsis thaliana* Col-0 and in leaf protoplasts of Col-0, *erf4* and *esp/esr* knock-out mutants. They were transfected with plasmid DNA purified with the Gene JET Plasmid Midiprep kit (Thermo Scientific) for GUS Assays or the Nucleo-Bond Xtra Midi kit (Macherey-Nagel, Düren, Germany) for BiFC. Protoplasts were transiently transformed with different concentrations of the respective plasmid DNA following the protocol published in Reference [36]. For details, see the protocol on http://www.zmbp.uni-tuebingen.de/c-facilit/plant-transformation.html.

### 2.7. GUS Reporter-Gene Assay in Protoplasts

For the GUS assays, 1.8 kbp upstream of the start codon of *CAT3* and 2.8 kbp upstream of the start codon of *WRKY53* were cloned into the GUS reporter plasmid pBGWFS7.0, respectively. The coding sequences of ERF4-A and ERF4-R [27] were cloned in the binary plant transformation vector pJAN33. Arabidopsis protoplasts were transformed as previously described with 3 μg effector constructs, 2.5 μg reporter plasmid and 0.1 μg of a *35S*:*Luciferase* plasmid (pBT8-35SLUCm3) for normalization purposes. The protoplasts were incubated over night at room temperature and the GUS assay was performed the next morning as described by Jefferson et al. [37].

### 2.8. BiFC and Flow Cytometry

For the BiFC assay, the coding sequence of *ERF4-R* and *ERF4-A* as well as *WRKY53* (as a negative control with ERF4 isoforms and as positive control with itself) were cloned into the BiFC-2in1-NN vector [38]. It fuses the N- and C-terminal part of eYFP to the N-termini of both interaction partners and it contains RFP (red fluorescent protein) as a transformation control. Protoplasts were transfected with 3 μg of the plasmid, covering the 8 possible combinations between the potential interaction partners and the negative control. After incubation overnight, fluorescence was quantified by flow cytometry using a CytoFLEX (Beckman Coulter, Brea, CA, USA). Both the internal mRFP and any reconstituted eYFP were excited by the on board 488 nm laser. For eYFP, peak emission was captured in FL1 (525/40 nm) and for RFP in FL3 (610/20 nm). After compensation, the percentage of cells with eYFP signal was determined. The results were means from 7 experiments for the ERF4 homo- and heterodimers and 3 experiments for the combinations with WRKY53. Furthermore, the interactions were visualized with a Leica TCS SP8 microscope. eYFP was excited at 514 nm and the emission was recorded at 519–560 nm, whereas mRFP was excited at 561 nm and the emission was recorded at 566–637 nm.

### 2.9. Plant Material

*Arabidopsis thaliana* plants were grown in standard soil under long-day conditions (16 h day, 8 h night) at an ambient temperature of 20 °C with a moderate light intensity (80–100 μE/m^2^/s). Individual leaves were labelled with differently colored threads according to their age. This allowed the identification of the leaf numbers within the first rosette even in very late stages of senescence [39]. Plants were always harvested at the same time in the morning in order to avoid circadian effects. The T-DNA insertion lines of *ERF4* (SALK_073394), *FPA* (SAIL720_B10, SALK_011615, SAIL_849_F10) and *ESP/ESR* (SALK_055029C) were obtained from the Nottingham Arabidopsis Stock Centre (NASC). Homozygous plants were verified by PCR using combinations of gene specific primers and the T-DNA left border primer (Lba1). Plants overexpressing the different isoforms of ERF4, as well as complementation lines were kindly provided by G. Simpson and K. Shirasu [27]. Different plant lines overexpressing ERF4-R isoform were obtained from and published by T. Koyama [17].

### 2.10. Semi-quantitative-PCR (sqRT-PCR) and qRT-PCR

Total RNA was extracted with the RNeasy plant mini kit (Qiagen) and the subsequent cDNA synthesis was done with RevertAid reverse transcriptase (Thermo Scientific) and oligo (dT) primer using 200 μg of RNA. qRT-PCR was performed using Kapa SYBR® Fast Bio-rad iCycler (Kapa Biosystems, Wilmington, MA, USA) master mix following the manufacturer’s instructions in a final volume of 8 μL including 3.5 μL of 1:5 diluted cDNA. The expression of the genes analyzed was normalized to *ACTIN2* (*ACT2*) according to the method published by Pfaffl [40]. sqRT-PCR products were amplified in 10 μL PCR reactions containing 1 μL of cDNA for amplification of *ACT2*, 2 μL for *ERF4-R* and 3 μL for *ERF4-A*. Red Mastermix (2x) (Genaxxon bioscience, Ulm, Germany) was used with 0.5 μM primer P1, P2 or P3 (see Appendix A). The exponential range of amplification was determined for each set of primers and accordingly different numbers of cycles were used (26 for *ACT2*, 27 for *ERF4-R* and 36 for *ERF4-A*) with different annealing temperatures (55 °C for *ACT2*, 62 °C for *ERF4*-R and 64 °C for *ERF4-R*; Appendix A). PCR products were separated on a 1% agarose gel with subsequent ethidium bromide staining and band intensity was analyzed using ImageJ.

### 2.11. Senescence Phenotyping

To assess differences in the progression of senescence between the plant lines, we measured a variety of parameters, which indicate the state of senescence from 25 up to 70 days after sowing (DAS). All methods are described in detail by Bresson et al. [39]. Electrolyte leakage was measured using a conductivity meter (*CM100-2,* Reid and Associates). In order to measure the hydrogen peroxide (H_2_O_2_) level, 3,3’-diaminobenzidine (DAB) staining or the fluorescent dye H2DCFDA were used. The activity of photosystem II (PSII) was assessed by Fv/Fm values (using the Imaging- PAM chlorophyll fluorometer Maxi version; ver. 2-46i, Walz GmbH, Effeltrich, Germany) and the status of chlorophyll breakdown was measured by extraction of chlorophyll (Chl). Leaf No. 6 was used for native catalase zymograms, RNA was extracted from leaves No. 7 and 8 and the leaf color was assessed with the automated colorimetric assay (ACA) published in Bresson et al. [39]. Chl content in the ERF4 complementation lines was measured with an *atLEAF+* Chl meter in leaf No. 5, three times per leaf and the values were averaged. All phenotyping experiments were done with a minimum of four biological replicates and experiments were performed at least three times independently.

### 2.12. Catalase Zymograms and Immunodetection

Plant protein was extracted out of frozen leaves by grinding material in 200 μL extraction buffer (100 mM Tris, 20% glycerol (v/v), 30 mM DTT, pH 8). The samples were centrifuged for 15 min at 14,000 rpm and 4 °C. Total protein concentration of the supernatant was measured via Bradford assay (Bio-Rad) and 10 μg total protein was loaded on native polyacrylamide (PAA) gels (6% PAA, 1 M Tris, pH 6.8; running buffer: 25 mM Tris, 250 mM Glycine, pH 8.3). After protein separation, the gels were rinsed twice with water, incubated for 2 min in a 0.01% H_2_O_2_ solution and rinsed again twice with water. The staining was performed in a solution containing 1% FeCl_3_ and 1% K_3_(Fe(CN)_6_) (w/v) until catalase activity bands became visible (~3 min). The reaction was stopped by rinsing gels with water. For immunodetection the native PAA gel was blotted on a nitrocellulose membrane. After blotting, the membrane was washed twice with Tris-buffered saline (TBS) and blocked with 3% milk powder in TBS-Tween 20 (TBS-T). Polyclonal anti-rye-CAT antibodies in 1.5% milk powder were used, followed by secondary peroxidase-conjugated antibodies for visualization.

## 3. Results

### 3.1. Impact of ERF-A and ERF-R on Senescence

*ERF4* is a transcription factor, which can undergo alternative polyadenylation resulting in the co-existence of two different isoforms, ERF-A and ERF-R [27]. We were interested to find out when both forms are produced during plant development and whether or not they have an impact on senescence. Therefore, we used an *erf4* mutant line, an *ERF4-A* and different *ERF4-R* overexpressing lines all in Col-0 background. The plants were grown side by side with Col-0 wildtype plants to compare senescence onset and progression. These phenotyping experiments were repeated three times with similar results. In order to investigate leaf senescence of the different plant lines, optical appearance was scored for several parameters during leaf development at the same rosette position from 25 to 70 days after sowing (DAS). Therefore, the true leaves (without the cotyledons) were labelled with different threads following a color code. Leaves of one rosette were sorted according to their age and corresponding leaves of the different lines were compared (Figure 1A). A statistical analysis of at least five plants was used to capture variability between individual plants of the same lines. A colorimetric analysis consisted in categorizing leaves into four groups depending on their leaf color: fully green, green/yellow, fully yellow, brown/dry (Figure 1B). Compared to Col-0 the 35S:*ERF4-R* line showed more signs of senescence, whereas the *erf4* line was less senescent at the same time point according to visual features (Figure 1A,B) indicating that the 35S:*ERF4-R* plants were accelerated and the *erf4* plants delayed in senescence. The 35S:*ERF4-A* plants behaved intermediately; at 41 DAS they were slightly accelerated whereas from 46 DAS onward, they appeared to be more and more delayed. The photosynthetic capacity was analyzed by chlorophyll fluorescence measurements of leaf No. 5 (Figure 1C). Consistent with the colorimetric analysis, the activity of photosystem II declined earlier in 35S:*ERF4-R* leaves compared to Col-0 leaves, whereas this became more obvious in *erf4* mutant or *ERF4-A* overexpressing leaves much later. Membrane integrity was monitored by measuring electrolyte leakage (EL) of leaf No. 3 (Figure 1D). 35S:*ERF4-R* line displayed higher membrane deterioration at 37 DAS compared to Col-0. In very late stages (53 DAS), Col-0 and 35S:*ERF4-R* plants did not show any more differences, whereas *erf4* and 35S:*ERF4-A* had higher membrane integrity compared to wild type. Intracellular ROS production was measured in leaf No. 9 by H_2_DCF-DA fluorescence (Figure 1E) and diaminobenzidine (DAB)-staining (Figure 1F). A significantly lower increase of intracellular H_2_O_2_ was detected in *erf4* mutant plants with advancing senescence compared to wildtype plants. Again, the 35S:*ERF4-A* overexpressing plants were more similar to the mutant line, whereas the 35S:*ERF4-R* line showed a faster increase in H_2_O_2_ content supporting the results of the colorimetric analysis (Figure 1A).

For comparison of the different lines at the same time point, it is important that their general development is not significantly altered. None of the lines showed a severe impairment in development (Appendix A). The *ERF4-R* overexpression lines showed a slightly delayed flowering and silique emergence (Appendix A) and had more smaller, less dense leaves (Appendix A) whereas *erf4* plants had a bigger leaf area and were heavier compared to wildtype plants (Appendix A). These minor changes in overall development still allow comparison of the lines.

Gene expression of senescence-associated genes (SAGs) was analyzed by quantitative real-time PCR using leaf No. 7 (Appendix A). The lower transcript levels of the senescence-upregulated short chain alcohol dehydrogenase *SAG13* (At2g29350) suggest that the *erf4* and the 35S:*ERF4-A* line are both delayed in senescence-associated gene expression compared to Col-0. The *ERF4-R* overexpression line showed a faster increase of *SAG13* mRNA levels until 46 DAS, then transcript levels in Col-0 reached higher values pointing to a slightly accelerated senescence compared to Col-0. *SAG12* (At5g45890), a senescence-upregulated cysteine protease that is often used as marker gene in late senescence, showed a different pattern for the later stages of leaf senescence. Col-0 and the mutant line showed the highest expression level, whereas the overexpression of *ERF4-A* resulted in the lowest expression of *SAG12* at 53 DAS. Down-regulation of *RUBISCO* (RBCS1) was not much altered in all lines indicating that ERF4 affects only part of the senescence reprogramming at different time points of senescence. 

Taken together, the results of the phenotyping experiments reveal that the *erf4* plants show delayed senescence, 35S:*ERF4-A* plants are only slightly delayed in senescence and more similar to Col-0, 35S:*ERF4-R* plants show an accelerated senescence phenotype, more pronounced in earlier stages and then becoming progressively indistinguishable to Col-0 plants in later senescence stages.

### 3.2. ERF4-A mRNA is Less Abundant than ERF4-R, but the Protein is Less Prone to Degradation In Vitro

In order to get an idea about the abundance of the *ERF4* isoforms at different stages of leaf senescence, we analyzed the relative amount of the mRNA forms in Arabidopsis Col-0 plants from 31 to 46 DAS by conventional RT-PCR as described by Lyons et al. [27]. Three different mRNA molecules can be derived from *ERF4* transcription: *ERF4-R*, *ERF4-IR* and *ERF4-A* as illustrated in Figure 2A. *ERF4-R* is created by using the first polyadenylation site, *ERF4-IR* by using the second polyadenylation site without splicing of an intron, which now exists in the longer mRNA. Therefore, this form is called IR for intron retention. *ERF4-A* is also created by using the second polyadenylation site, however, in this form the intron is spliced out, which now leads to the loss of the repressing EAR domain. Therefore, this form is called A for activator. These three mRNA forms lead to two different protein variants whereby *ERF4-A* mRNA encodes the ERF4-A protein and *ERF4-R* and *ERF4-IR* mRNAs both lead to the ERF4-R protein. The nucleotide sequence of *ERF4-IR* entirely includes the sequence of *ERF4-R* and as well as of *ERF4-A*. This is the reason why no sequence specific primers can be designed to distinguish all three forms by quantitative RT-PCR. Therefore, a semi-quantitative RT-PCR approach was chosen using two different primer pairs: P1 and P3 differentiate the *ERF4-IR* form from the *ERF4-A* form by size and P1 and P2 amplify the *ERF4-R+IR*. For simplicity, we called the sum of *ERF4R+IR* only *ERF4-R* in the following as they encode the same protein. First, we determined the exponential range of amplification (Appendix A). Thereafter, we used 36 cycles and 3 μL cDNA for the *ERF4-A* and *ERF4-IR* amplification, 27 cycles and 2 μL cDNA for *ERF4-R* and 27 cycles and 1 μL of cDNA for *ACTIN2* as reference gene. These initial results indicated that the A-Form is much less abundant than the R-Form. The *ERF4-IR* mRNAs were even less abundant than the *ERF4-A* form and, since they code for the same protein as *ERF4-R,* were not analyzed separately. From these analyses, we determined that the *ERF4-A* and *ERF4-R* isoforms show an age-dependent expression, with a higher mRNA abundance in later stages of leaf senescence (Figure 2B). However, after bolting at 31 DAS, a decrease in the expression of *ERF4-A* can be observed, followed by a constant increase, indicating that both isoforms are not always produced at the same ratio over development (Figure 2C). Moreover, the expression of the different isoforms was also analyzed in the 35S:*ERF4-R* and 35S:*ERF4-A* as well as in the *erf4* mutant line. (Appendix A). Both isoforms were clearly overexpressed and overexpression of the *ERF4-R* led to a slight decrease of *ERF4-A* formation and *vice versa*. However, how this feedback loop on splicing and polyadenylation is achieved, has to be further investigated.

To investigate whether the lower abundance of ERF4-A is counterbalanced by a higher protein stability, we subjected recombinant proteins of both isoforms to in vitro proteolysis by plant extracts. Therefore, epitope-tagged versions of the proteins were expressed in *E. coli* and 25 μg of the protein crude lysates containing the recombinant proteins were incubated for 0, 15, 30 and 60 min with Arabidopsis leaf extracts, which were derived from selected rosette leaves (9–13 or 7–8) at 49 DAS. The ERF protein amount, which remained after incubation, was assessed by immunodetection. Both ERF4 isoforms were rapidly degraded by the plant extracts (Figure 2D), whereas ERF4 protein amounts were not reduced under the same conditions by bacterial cell lysates (Appendix A), suggesting that ERF4 isoforms are degraded by plant-specific proteins and/or the plants’ proteasome. Both isoforms were more stable in extracts of younger leaves (No. 9–13) than in extracts of older leaves (No. 7–8); ERF4-A was in general more stable than ERF4-R. This suggests that both ERF4 isoforms are present throughout the whole process of leaf senescence, lower expression in younger leaves is counterbalanced by higher protein stability in this tissue, and, in addition, lower *ERF4-A* mRNA amounts are compensated by a higher protein stability of ERF4-A compared to ERF4-R.

### 3.3. Senescence Analyses of erf4 Complementation Lines

To further analyze the impact of each isoform on senescence, we characterized complementation lines, which express the *ERF4-R* or *ERF4-A* coding sequence under the control of the native *ERF4* promoter in the *erf4* background (cERF4-R and cERF4-A). These lines were obtained from Lyons et al. [27]. After verification of the isoforms’ expression by semi-quantitative PCR (Appendix A), the plants were grown along with Col-0 and the *erf4* plants under long-day conditions and low light intensities (30 μE/m^2^/s) to slow down development and increase the resolution of senescence. As before, leaves were sorted according to their age and one example (55 DAS) is presented in Figure 3B. As the plants were grown under low light conditions, Col-0 leaves were less senescent at 55 DAS compared to Figure 1A,B at 53 DAS. The differences between the complementation lines were very subtle. In order to quantify leaf color more precisely than by eye, we used an automated colorimetric assay (ACA) to classify the color values pixel-wise (Figure 3A, [39]). According to the ACA, both lines were able to complement the mutation of *ERF4* almost completely with a tendency of cERF4-A to complement slightly better (Figure 3A,B).

DAB-staining indicated that H_2_O_2_ levels in *erf4* plants are lower compared to Col-0 levels as the plants age. Both, *ERF4-A* and *ERF4-R,* complemented the *erf4* mutant, with *ERF-A* again slighly more effective than the *ERF-R* form, as staining of cERF4-A leaves appear to be more similar to wildtype than to cERF4-R leaves (Figure 3C). Fv/Fm levels of cERF4-A resembled those of wildtype plants and decreased earlier than cERF4-R and the mutant line; however, from 55 DAS onwards both complementation lines complemented equally well (Figure 3D). With regard to EL in leaf No. 4, the cERF4-A line also shows the tendency of complementing *ERF4* loss slightly better than cERF4-R (Figure 3E). In summary, both isoforms can complement the loss of ERF4 in *erf4* plants almost to wildtype level with a tendency of the *ERF4-A* for a slightly better complementation. This is very surprising as Lyons et al. [27] have already shown that both forms can act antagonistically on gene expression, at least on *PDF1.2*.

### 3.4. Impact of ERF4 on the Senescence Regulator WRKY53

As shown by Koyama and coworkers [17], WRKY genes are significantly upregulated in *ERF4-R* overexpressing lines and *WRKY53* senescence-associated induction is eliminated in the *erf4/erf8* double mutant. The authors explained the effect by the action of ESP/ESR, which is a negative regulator of *WRKY53* expression and activity [18]. By ChIP, *ESP/ESR* was shown to be a direct target of ERF4 acting as repressor on *ESP/ESR* expression [17]. In order to investigate whether there is also a direct effect of both ERF4 isoforms on the expression of *WRKY53*, we transiently co-transformed a 2.8 kbp *WRKY53*-promoter:*GUS* construct with 35S:*ERF4-R* and 35S:*ERF4-A* effector constructs in Arabidopsis Col-0 or *esp/esr* protoplasts. In the presence of ESP/ESR, *WRKY53* expression was increased approx. 5-fold by *ERF4-R* overexpression (Figure 4), which is in the same range as in the microarray experiment of Koyama et al. [17]. However, in the absence of ESP/ESR in *esp/esr* protoplasts, *WRKY53* expression was induced approx. 2-fold by *ERF4-R* overexpression, indicating that *WRKY53* expression is also influenced in an ESP/ESR-independent pathway (Figure 4). Overexpression of ERF4-A showed a weak but positive effect (approx. 1.5-fold) on this promoter, which was almost completely missing in the *esr/esp* protoplasts (Figure 4) indicating that ERF4-A has only a weak effect but uses exclusively the ESP/ESR pathway.

### 3.5. Isolation and Characterization of CATALASE3 (CAT3) as Direct Target Gene of ERF4

In a yeast-one-hybid (Y1H) screen, which was performed to find potential regulatory proteins that interact with the *CAT3* promoter, different *CAT3* promoter fragments were cloned in front of reporter genes (*His* and *Ade*, *LacZ*) and brought into yeast cells together with a cDNA library obtained from 7-week-old Arabidopsis leaves. cDNA insertions were sequenced and identified from yeast cells growing on selection media. Among others, partial sequences of *ERF4* were identified in two independent clones to bind to a 150-bp fragment (−332 to −182) of the *CAT3* promoter (Figure 5A). Therefore, the full-length coding sequence of *ERF4-R*, which is the predominantly expressed isoform, was cloned into the pGADT7 vector and transformed into the reporter yeast strain Y187 [pHISi-*CAT3*] and growth on selection media could be confirmed. Therefore, the full-length ERF4-R protein binds to a *cis*-element in the *CAT3* promoter. The DNA-binding of ERF4 to the *CAT3* promoter was further analyzed in vitro by DPI-ELISA. For DPI-ELISA, the coding sequence of both *ERF4-R* and *ERF4-A* isoforms were cloned in pETG10A with N-terminal hexa-histidine-tag (6xHIS-tag) and expressed in bacteria. A 55-bp biotinylated DNA fragment of the *CAT3* promoter sequence, which was identified in the yeast-one-hybrid screen and contains the DRE motif CAGCC, an already known binding motif for ERF4 [41], was used (Figure 5A). Whereas the crude lysate of the *Escherichia coli* (*E. coli)* strain expressing the empty vector did not show any binding activity, both ERF4 isoforms were able to bind to this fragment (Figure 5B).

To analyze the effect of ERF4 isoforms on the expression of *CAT3* in vivo, transient co-transformations of a 1.8 kbp *CAT3*-promoter:*GUS* construct along with 35S:*ERF4-R* or 35S:*ERF4-A* overexpression constructs in Arabidopsis Col-0 or *erf4* protoplasts were analyzed for GUS protein expression (Figure 5C). Protein formation of ERF4-R and ERF4-A was identified in the transformed protoplasts by immunodetection (Appendix A). Reporter gene expression was repressed by the overexpression of *ERF4-R* as expected for a direct target gene of a protein containing an EAR motif, whereas overexpression of *ERF4-A* resulted in an increased reporter gene activity (Figure 5C). This clearly suggests that the isoforms have an antagonistic function in the regulation of *CAT3*. Simultaneous co-transformation of both isoforms led to reporter gene repression (Figure 5C) suggesting that the repressor isoform is dominant over the activator form. 

CAT3 protein amount and activity increases during development and progression of senescence in wildtype plants (Appendix A; [7,42]). When the activity of the catalase isoforms was analyzed by zymograms in Col-0 and *erf4* mutant plants during plant development, a peculiar activity pattern could be observed for CAT3. As band intensities on catalase zymograms are proportional to enzyme activity in a wide range [7], CAT3 activity was quantified using ImageJ. In 5-week-old plants, CAT3 activity was lower in *erf4* leaves compared to leaves of Col-0 plants, which is more obvious in the growth series shown in Appendix A, most likely due to slightly different velocities in development of different growth series. In 6-week-old plants, CAT3 activity was equal in *erf4* plants compared to wildtype and in 7- and 8-week-old plants CAT3 activity is higher than in Col-0 plants (Figure 5D; Appendix A). This suggests that the loss of ERF4 has an age-dependent influence on CAT3 activity: at early stages the activator ERF4-A appears to have a higher impact whereas at later stages of development the repressor ERF-R has a dominant effect.

### 3.6. ERF4 Isoforms Form Homo- and Heterodimers, but Homodimers are Less Favored

The existence of the two different isoforms raised the question, whether or not they form homo- and/or heterodimers. Therefore, we first analyzed the formation of different dimers in vitro by a pull-down of recombinant proteins via Ni-agarose-beads. *E. coli* BL21 Rosetta cells were transformed with constructs coding for N-terminal GST-tagged and N-terminal 6xHIS-tagged fusion proteins (Figure 6A). GST-ERF4-A and GST-ERF4-R could be successfully pulled down with both HIS-ERF4-A and HIS-ERF4-R proteins. These results demonstrate that in vitro both the formation of homo- and heterodimers is possible. 

To confirm the dimerization in planta, bimolecular fluorescence complementation (BiFC) combined with flow cytometry was performed. Arabidopsis Col-0 protoplasts were transformed with constructs coding for the sequences of *ERF4-A* and *ERF4-R*, which were N-terminally fused to the sequences coding for the N- and C-terminal part of eYFP, respectively. As positive control *WRKY53* constructs were included, since WRKY53 proteins are already shown to form homodimers [23]. Flow cytometry results show that homodimers of both ERF4 isoforms could be formed with the same probability as WRKY53 homodimers (Figure 6B,C and Appendix A). In contrast to the pull-down experiments, heterodimers between both ERF4 isoforms and with WRKY53 are—if at all—formed with a much lower probability. They are not significantly different from the background signals of the mRFP control. Under the confocal laser-scanning microscope, the YFP signal of the BiFC is predominantly localized in the nucleus (Appendix A).

### 3.7. FPA Inhibits ERF4-A Formation and Appears to be Involved in Senescence Regulation

Lyons and coworkers observed that the RNA-binding protein FPA inhibits the formation of the ERF4-A form by promoting the polyadenylation in favor of the ERF4-R form [27]. We could show by qRT-PCR that *FPA* exhibited a senescence-specific expression pattern with rising expression levels in older plants (Figure 7A) that coincide with a decrease in *ERF4A/ERF4R* ratio (Figure 7B). In *erf4* plants, the increase of FPA expression is delayed most likely due to the delayed senescence of *erf4*. Furthermore, sqRT-PCR confirmed that *ERF4-A* accumulates to much higher expression levels in *fpa* mutant plants (Figure 7C) as observed previously by Lyons et al. [27]. Therefore, we wanted to analyze whether FPA also has an influence on leaf senescence and therefore characterized two *fpa* T-DNA insertion lines (SAIL 849-F10 and SAIL 720-B10) in that regard. Each T-DNA insertion was confirmed by PCR. As FPA also controls flowering by repressing *FLOWERING LOCUS C* expression, the mutant plants have a strong delay in bolting and flowering. The shoot apical meristem (SAM) developed 30 and 40 leaves before the plant started bolting, as compared to Col-0, which had around 13 leaves (Figure 7D). Therefore, senescence phenotyping is very difficult as ideally plants of the same developmental stage should be compared. Nevertheless, with regard to the first 13 leaves, *fpa* mutants displayed accelerated senescence (Figure 7E) possibly indicating that FPA is not only involved in regulation of flowering time but also in senescence. 

## 4. Discussion

Leaf senescence goes along with massive changes in gene expression [12] and transcription factors play a key role in the differential expression of genes. Class II ERFs are transcription factors involved in many different processes and signaling pathways. They participate in the responses to pathogen, ethylene, SA, JA, auxin, salt and ABA [43,44,45,46,47,48]. Moreover, they are involved in cell death and senescence processes. For example, NbCD1, a *Nicotiana benthamiana* class II ERF, has been characterized to positively regulate cell death [49] and the AP2/ERF factor MACD1 as well as ERF102 were also reported to be involved in phytotoxin-triggered programmed cell death in *Nicotiana umbratica* [50]. Furthermore, *Nt*ERF3 and other ERFs containing the EAR motif of tobacco or rice were able to induce hypersensitive response (HR) when overexpressed in plants [51]. Koyama and coworkers [17] identified a regulatory cascade involving Arabidopsis ERF4 and ERF8 to promote leaf senescence. Even though the growth regulator ethylene is part of its name, responsiveness to this phytohormone is not a collective feature of all ERFs. Moreover, ERFs can have dual functions, as for example ERF-VII proteins act as positive regulators of the hypoxic response and as repressors of oxidative-stress related genes, depending on the developmental stage of the *A. thaliana* plants [52]. In this study, we expand the context and/or age-dependent function of ERFs to ERF4 of Arabidopsis and describe the role of two ERF4 isoforms derived from alternative polyadenylation in senescence. Phenotyping experiments show that both isoforms have a different and opposing impact on senescence: whereas 35S:*ERF4-R* plants show an accelerated phenotype, 35S:*ERF4-A* plants are slightly delayed in leaf senescence (Figure 1A–F). Complementation of the ERF4 loss in *erf4* mutant plants indicated that both isoforms were able to complement the delayed senescence of *erf4* with a tendency of ERF4-A to restore the wildtype phenotype slightly better (Figure 3A–E). This indicates that ERF4-A and ERF4-R both contribute to the senescence regulatory effect of ERF4 having partially overlapping functions and that alternative polyadenylation is part of the regulatory network of senescence. Nevertheless, it was unexpected that both isoforms were able to complement the mutant phenotype since both isoforms exhibit opposing effects on the expression of the direct target gene *CAT3* (Figure 5C) and *PDF1.2* [27]. However, *WRKY53* expression was affected in the same direction by both isoforms, either directly or indirectly. Microarray analyses of the *ERF4-R* overexpressing line revealed that not only *WRKY53* but also many other WRKY transcription factors are downstream targets of ERF4 [17] and many of those are involved in senescence regulation. Moreover, WRKYs act in a transcriptional network influencing the expression of each other so that the outcome on senescence can hardly be predicted [21,23]. On the other hand, single knock-out mutants of transcription factors involved in senescence regulation often do not show any obvious phenotype even though they are responsible for changes in the transcriptome. These changes in gene expression are contributing to the senescence process as a whole but do not necessarily relate to a visible phenotype. Therefore, complementing only one of the two isoforms might be sufficient to restore the phenotype even though the absence of the other still leads to changes in senescence-associated genes (SAG) expression, which could explain that both isoforms were able to complement the mutant phenotype. Therefore, a general transcriptome analysis of the complementation lines will be subject of further investigations and will give insight in differential changes to identify specific target genes of either isoform. Both complementation lines performed slightly different in complementing the mutant: ERF4-A appears to be more important at earlier stages of senescence and more involved in regulating intracellular H_2_O_2_ concentration (Figure 3C) whereas ERF4-R appears to promote membrane integrity (Figure 3E). Therefore, it can be assumed that different and specific target genes and pathways are targeted by the different isoforms. The simultaneous presence of the two opposing isoforms render *in planta* function of ERF4 very complex. The total amount of the *ERF4* mRNA increases with age and both isoforms can be detected in leaves throughout development from 31-46 DAS. However, the ratio between the mRNAs of the two isoforms changes over development. The *ERF4-A* form is always less abundant than the *ERF4-R* form, but at 31 DAS the A:R ratio is highest (Figure 2C). To counterbalance the low abundance of the *ERF-A* mRNA, the resulting ERF-A protein is more stable than the ERF4-R protein (Figure 2D). 

The formation of the *ERF4-A* form was shown to be dependent on FPA, a RNA-binding protein, which inhibits the alternative polyadenylation [27]. Expression of *FPA* is dependent on plant age and increases with senescence (Figure 7A). Consistent with this increase, the ratio between *ERF4-A* and *ERF4-R* is lower at a later time point (Figure 7B). This increase in *FPA* expression is less pronounced in *erf4* mutant plants (Figure 7A) due to either delayed senescence or a feedback regulation of ERF4 on *FPA*. As FPA also controls flowering by repressing *FLOWERING LOCUS C* expression, the mutants have a strong delay in bolting and flowering. Therefore, it is difficult to compare senescence with Col-0 plants, as, in general, plants of the same developmental stage should be compared. Whereas Col-0 plants developed only 13 leaves and had already flowers and siliques, *fpa* mutants continued to produce more than 30 leaves without bolting and flowering. However, if the first 13 leaves of plants with the same age are compared, *fpa* mutants might be accelerated in senescence in these leaves. As *fpa* plants have higher amounts of ERF4-A (Figure 7C), the phenotype should resemble the 35S:*ERF4-A* overexpressing line. However, this line shows only a slightly accelerated senescence at 41 DAS, and at later time points it appears to be slightly delayed, indicating that this needs further investigations and that FPA is most likely involved in the regulation of polyadenylation of more target genes. Remarkably, FPA is involved in the regulation of two processes, which show overlapping gene expression patterns with senescence, namely flowering and pathogen response. This suggests that alternative polyadenylation is another mean for the cross-talk between these three processes.

We identified ERF4 as a DNA-binding protein of the *CAT3* promoter in an Y1H screen (Figure 5A). Some members of the ERF family have been shown to bind in vitro to an AGCCGCC motif (GCC-box) [53], while other ERFs have also been reported to bind DRE elements [41], or even novel DNA elements [54,55]. We confirmed the in vitro binding of both protein isoforms ERF4-R and ERF4-A to the *CAT3* promoter to a fragment containing the DRE motif with DPI-ELISA (Figure 5B). Reporter gene assays revealed that ERF4-R containing the EAR motif always had a repressing effect on reporter gene expression controlled by the *CAT3* promoter, whereas ERF4-A lacking the EAR motif activated *CAT3* driven reporter gene expression in leaf protoplasts (Figure 5C). The *ERF4* gene also contains a conserved motif with high number of acidic residues, which might function as activating domain in the absence of the EAR motif (Appendix A). Our results are consistent with that of Lyons and coworkers [27] who could also show the transcriptional activation of ERF4-A on a GAL4-*GCC* promoter and the *PDF1.2* expression. Co-transformation of both effectors led to a repression (Figure 5C). Therefore, ERF4-A can activate *CAT3* expression but, when both isoforms compete with each other, it appears to be less efficient. 

*CAT3* is an interesting target gene for a senescence-regulating transcription factor, as catalases control H_2_O_2_ levels, which have been shown to act as signaling molecules in senescence. In planta, CAT3 activity increased with senescence in Col-0 plants (Figure 5D, Appendix A). In contrast, a complex activity pattern became obvious in the *erf4* mutant. In younger plants, activity is lower compared to wildtype indicating the loss of a CAT3 activating factor, whereas in older plants CAT3 activity is higher compared to Col-0 plants pointing at the loss of a repressing factor. This suggests that the ERF4-A isoform has its highest impact at an early time point of senescence, whereas ERF4-R is more important at later stages. This is consistent with the inhibition of ERF4-A formation by FPA in later stages (Figure 7A–C). All three catalases in Arabidopsis display a senescence-associated activity pattern, in which *CAT2* is downregulated during early senescence, *CAT3* is upregulated during progression of senescence and *CAT1* is upregulated during late senescence. A fine-tuned regulatory loop between CAT2, CAT3, and ASCORBATE PEROXIDASE1 activities leads to an early peaking of H_2_O_2_ in senescence, which is used as a signal to induce senescence-associated gene expression [7,56]. Moreover, activity of many transcription factors is directly controlled by redox conditions [57]. Therefore, ERF4 could affect senescence progression also via its impact on the signaling molecule H_2_O_2_ through activation or repression of *CAT3*. *CAT3* expression is in general induced by high levels of H_2_O_2_; however, during progression of senescence, this responsivity is specifically blocked while other stress responses are still working [58]. ERF4-R might be involved in the disruption of this substrate induction to guarantee an increase in ROS during senescence, which is important to induce *SAG* expression and in later stages also for membrane deterioration and macromolecule breakdown.

Figure 8 illustrates our hypothesis of how ERF4 isoforms influence the progression of leaf senescence. *ERF4* transcription rate increases with leaf age but due to increasing FPA levels in later stages of development the ratio between ERF4-A and ERF4-R forms is changed by alternative polyadenylation. In early stages, ERF4-A is enriched since *FPA* expression is low. At this time point ERF4-A positively influences the expression of *CAT3* and thereby keeps H_2_O_2_ levels low. With onset and progression of senescence, FPA levels increase and ERF4-A production is minimized. ERF4-R now negatively regulates *CAT3* expression leading to an increase in H_2_O_2_ production. This increased intracellular H_2_O_2_ levels now induce the expression of the senescence regulator *WRKY53* and many other senescence-associated transcription factors and SAGs.

Moreover, ERF4-R directly and negatively regulated the expression of *ESP/ESR*, which in turn is a negative regulator of *WRKY53* expression and activity. However, *WRKY53* appears to be upregulated by ERF4 also in an ESP/ESR-independent pathway (Figure 4), but it is still unknown, whether this is an indirect or direct regulation. The latter appears to be likely, since our in silico analyses identified at least four ethylene responsive *cis*-elements and a DRE element in the *WRKY53* 1000 bp upstream promoter region. Taken together, our findings reveal that ERF4-A and ERF4-R both contribute to the senescence regulatory effect of ERF4 and that alternative polyadenylation adds a further layer of complexity to the regulatory network of senescence.

## Figures and Tables

**Figure 1 genes-10-00091-f001:**
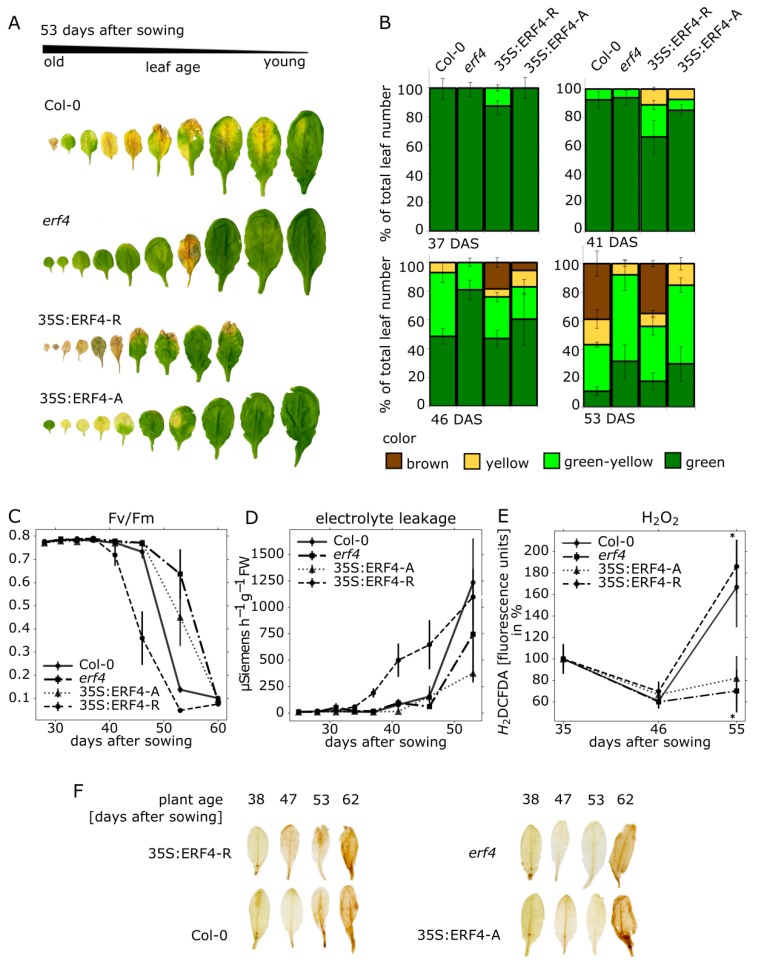
Senescence parameters analyzed in plant lines with altered *ERF4* expression. Col-0, 35S:*ERF4-R*, 35S:*ERF4-A* and *erf4* mutant plants were analyzed over development. (**A**) Representative pictures for rosette leaves No. 1–10, which were sorted according to their age 53 days after sowing (DAS). (**B**) Rosette leaves were categorized in four groups according to their color (fully green, green/yellow, fully yellow, brown/dry) and percentage of each category is depicted. (**C**) Fv/Fm values in leaf No. 5 were measured with the pulse amplitude modulation (PAM) method. (**D**) Electrolyte leakage (EL) over time was measured in single detached leaves No. 4 using a conductivity meter. Values are normalized to leaf fresh weight. (**E**) Intracellular H_2_O_2_ content per leaf was measured in leaf No. 9 using the fluorescent dye H_2_DCFDA. Values are normalized to leaf weight. Data represent means of 5 (A–D) or 10 (E) biological replicates, error bars present ±SE. (**F**) H_2_O_2_ content in leaf No. 9 is visualized by DAB (3,3-diaminobenzidine)-staining. One representative example is shown. Experiments were repeated three times with similar results.

**Figure 2 genes-10-00091-f002:**
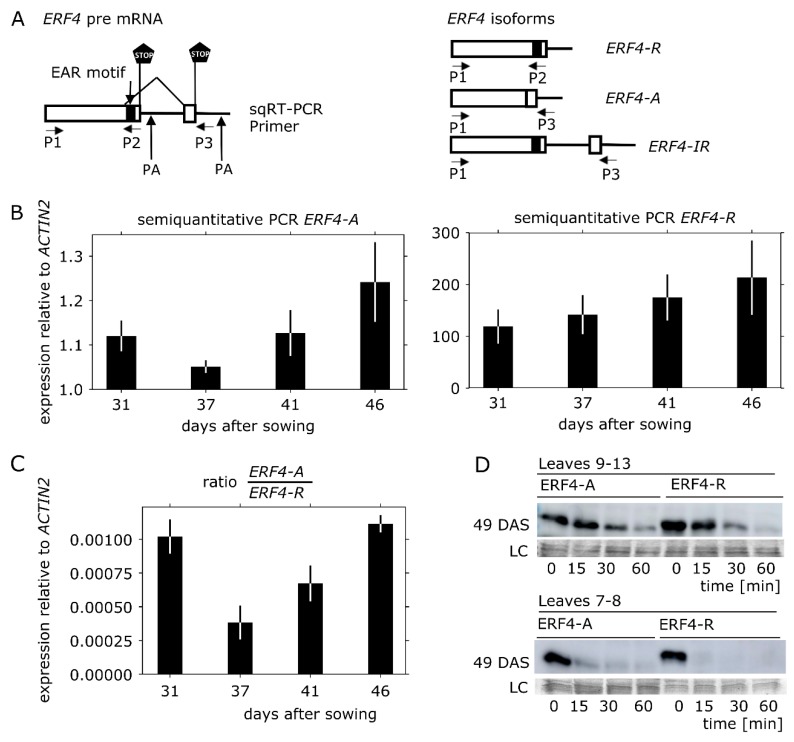
ERF4 isoform expression and protein stability. (**A**) Schematic illustration of *ERF4* pre-mRNA with two different polyadenylation sites and primer binding sites for primer 1 (P1), primer 2 (P2) and primer 3 (P3), which were used for semi-quantitative reverse transcriptase PCR (sqRT-PCR) as described in [27]. Boxes denote exons, lines denote introns and untranslated regions, PA indicates alternative polyadenylation sites, black box indicates the repressing ERF-associated amphiphilic repression (EAR) motif. cDNA of the *ERF4* isoforms is shown on the right side. (**B**) Results of the sqRT-PCR using RNA isolated from Col-0 plants at different developmental stages indicated in days after sowing (DAS). Graph shows expression of *ERF4-A* (36 cycles, 3 μL cDNA) and *ERF4-R* (27 cycles, 2 μL cDNA*)*, which was quantified using ImageJ and is presented relative to expression of *ACTIN2*. Error bars indicate ±SE, *n* = 3. (**C**) Ratio of *ERF4-A* to *ERF4-R* mRNA at different developmental stages (**D**) Cell-free protein degradation assay. 6xHIS-tagged ERF4-R (26.6 kDa) and ERF4-A (24.5 kDa) proteins were incubated in protein extract of young (No. 9−13) and middle old leaves (No. 7 and 8) for 0−60 min. Five plants were pooled for protein extraction and were harvested 49 days after sowing (DAS). Amido black staining of the upper part of the polyvinylidene difluoride (PVDF) membrane is shown as loading control (LC).

**Figure 3 genes-10-00091-f003:**
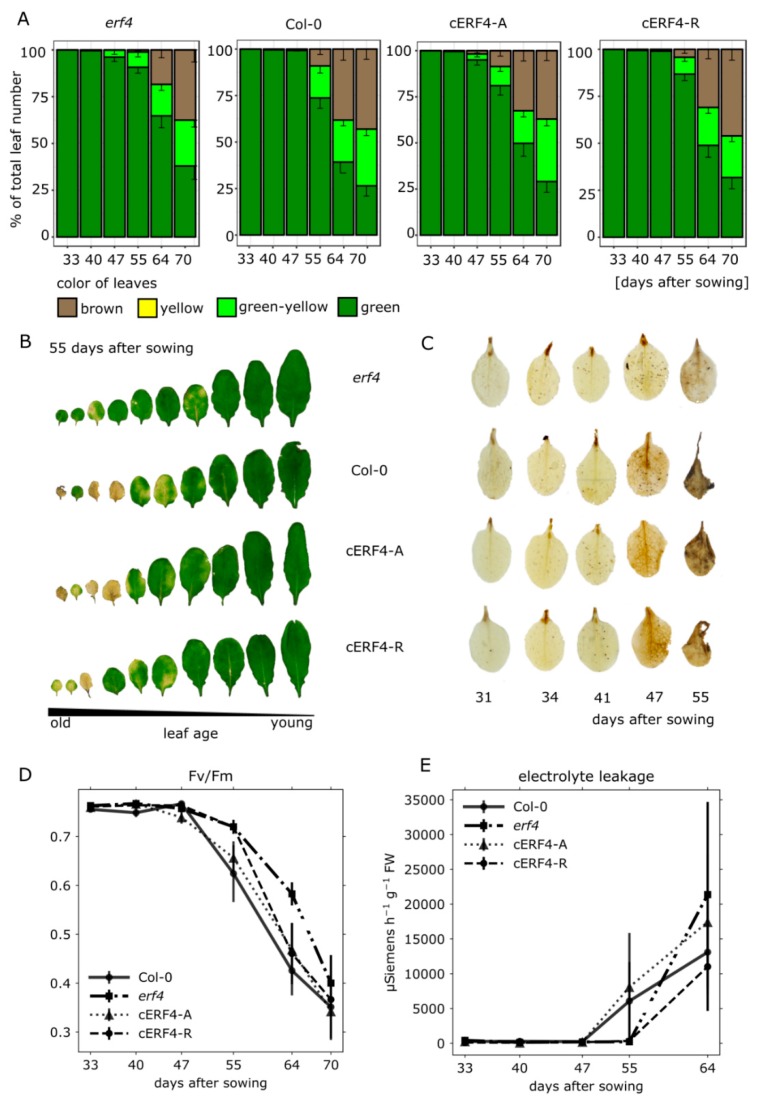
Complementation of ERF4 loss of function by ERF4-A or ERF4-R. Senescence phenotype and parameters were analyzed in Col-0, *erf4* mutants, and *erf4* mutant plants complemented with either *ERF4*-promoter:*ERF4-A* (cERF4-A) or *ERF4*-promoter:*ERF4-R* constructs (cERF4-R). (**A**) Automated colorimetric assay, in which rosette leaves are categorized according to color in four groups (green, green-yellow, yellow, brown). (**B**) Representative pictures of rosette leaves No. 1–10, which were sorted according to leaf age 55 DAS. (**C**) One representative picture of five biological replicates for DAB-staining in leaf No. 4 at 31–55 DAS. These plants have been raised under normal light intensities (80 μE/m2/s). (**D**) Fv/Fm values in rosette leaves were measured with pulse amplitude modulation (PAM). (**E**) Electrolyte Leakage (EL) was measured with a conductivity meter in detached leaves No. 4 of another experimental series. Values are normalized to fresh weight. Data A, D and E represent means of at least four biological replicates, error bars indicate ±SE.

**Figure 4 genes-10-00091-f004:**
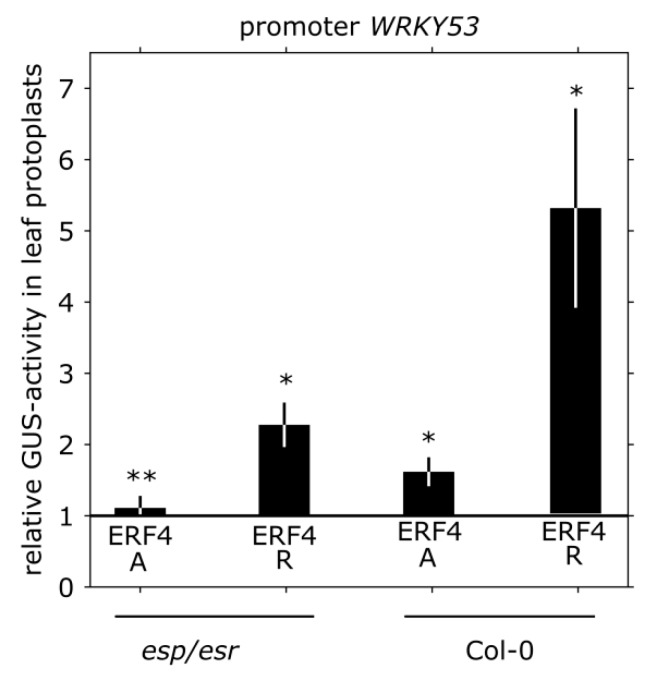
Impact of ERF4 on *WRKY53* expression. Transient expression assays were performed in Arabidopsis protoplasts with a 2.8 kbp fragment of the *WKRY53* promoter that drives expression of *GUS* in Col-0 and *esp/esr* mutant leaf protoplasts. Values represent fold-change of GUS-activity relative to the empty vector after normalization to luciferase activity from a minimum of four experiments. Error bars indicate ±SE. Statistical differences were determined by one sample t-test (* *p* ≤ 0.05, ** *p* ≤ 0.005, *** *p* ≤ 0.0005).

**Figure 5 genes-10-00091-f005:**
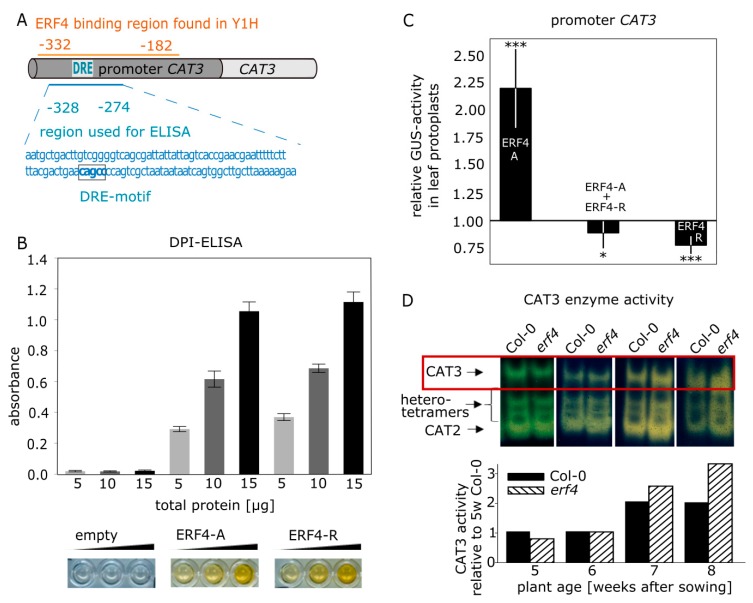
ERF4 binds to the *CATALASE3* promoter and can act as activator and repressor. (**A**) Schematic representation of the *CAT3* promoter fragments used for Y1H screen and for DPI-ELISA. (**B**) DPI-ELISA using bacterial crude extracts of *E. coli* BL21 Rosetta cells expressing 6xHis-ERF4-A, 6xHis-ERF4-R or the empty vector and a 55-bp biotinylated DNA fragment of the *CAT3* promoter, which contains the DRE motif CAGCC. Representative wells of the micro titer plate are shown below the graph. Yellow color indicates interaction. Error bars indicate ±SE, *n* = 4–5. (**C**) Transient expression assays were performed using the 1.8 kbp fragment of the *CAT3* promoter upstream of the start codon in Arabidopsis Col-0 and *erf4* leaf protoplasts. Values represent fold-change of GUS-activity relative to the empty vector after normalization to luciferase activity to compensate for differences in transformation efficiency. Data are means of seven experiments (*n* = 7). Statistical differences were analyzed by one sample t-test (* *p* ≤ 0.05, ** *p* ≤ 0.005, *** *p* ≤ 0.0005). (**D**) Enzyme activity of catalase isoforms of Col-0 and *erf4* mutant plants is visualized on a zymogram. Protein crude extract (10 μg) of leaf No. 6 of 5- to 8-week-old plants were separated on 7.5% native gels and stained for catalase activity. Proteins were extracted of a pool of at least 10 leaves. Intensities of the CAT3 bands were quantified using ImageJ.

**Figure 6 genes-10-00091-f006:**
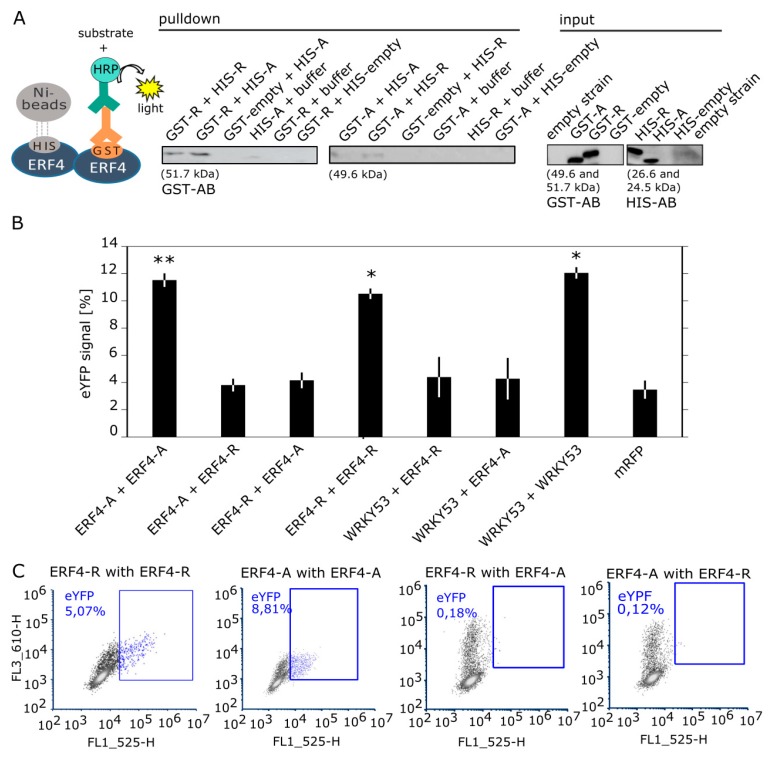
Protein–protein interaction of ERF4-A and ERF4-R.(**A**) Pull-down assay was performed using total protein extracts from *E. coli* BL21 Rosetta cells. GST-tagged ERF4-R (51.7 kDa) and ERF4-A (49.6 kDa) was pulled down with HIS-tagged ERF4-R (26.6 kDa) and ERF4-A (24.5 kDa) proteins and immunodetected with anti-GST antibodies. The input protein (25 μg) was visualized with anti-GST and anti-HIS antibodies. The experiment shown was performed three times with similar results. (**B**) BiFC flow cytometry experiments were performed in Arabidopsis protoplasts co-expressing ERF4-R and ERF4-A fused with YFP-N and YFP-C, respectively. YFP-N fusion with WRKY53 and YFP-C fusions with both ERF4 isoforms were used as negative controls and YFP-N and YFP-C fusions with WRKY53 as positive control. RFP is expressed as transfection control. Bars represent percentage of cells with eYFP signal (mean values (±SE), *n* = 7 for combinations with ERF4 isoforms, *n* = 3 for combinations with WRKY53). Data was subjected to quantile normalization and determination of statistical differences was carried out using Wilcoxon rank sum test (* *p* ≤ 0.05, ** *p* ≤ 0.005). (**C**) Representative graphs of the flow cytometry results for ERF4 homodimers and heterodimers. Blue dots represent eYFP signals of interaction. Blue squares mark the cells showing eYFP signal.

**Figure 7 genes-10-00091-f007:**
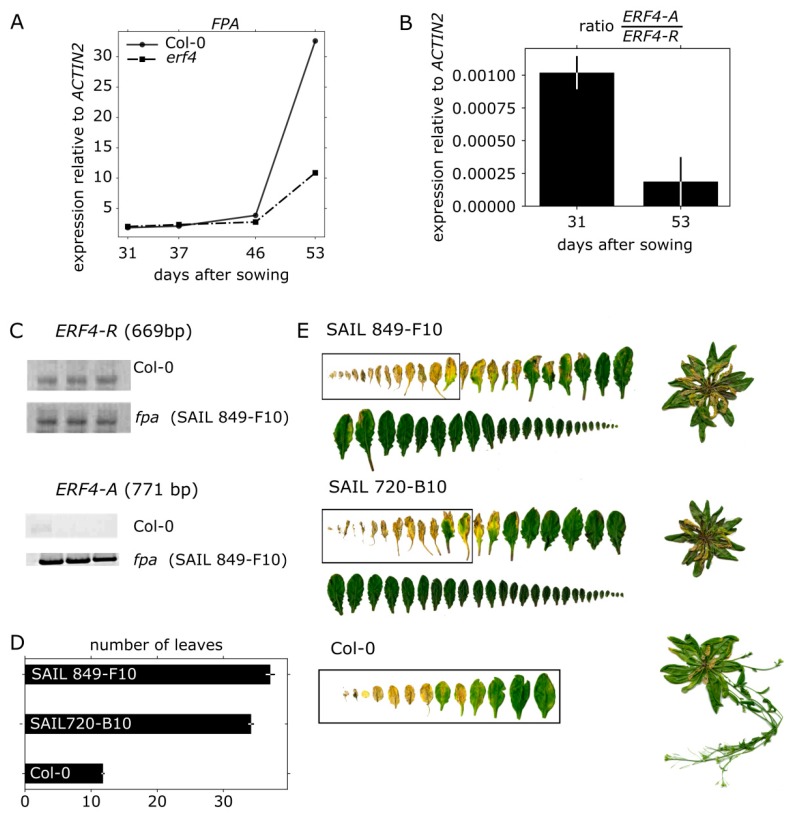
Characterization of *fpa* plants. (**A**) qRT-PCR data shows expression level of FPA in Col-0 and *erf4* plants relative to *ACTIN2*. Leaves No. 7 of 5 biological replicates were pooled. Data are means of two technical replicates. (**B**) Ratio between *ERF4-A* and *ERF4-R* mRNAs at early (31 DAS) and late stages of development (51 DAS) (**C**) Semi-quantitative-PCR (sqRT-PCR) with RNA isolated from a pool of 10 plants was performed. Images show representative bands on a 1% agarose gel for *ERF4-R* (27 cycles, 2 μL cDNA, 669 bp) and *ERF4-A* (36 cycles, 3 μL cDNA, 771 bp) in Col-0 and *fpa* (SAIL 849-F10) plants of three technical replicates. (**D**) Number of rosette leaves 6 weeks (42 DAS) after sowing of *fpa* and Col-0 plants. Error bars indicate ±SE, *n* = 10. (**E**) Representative pictures of rosette leaves sorted according to their age 7.5 weeks (51 DAS) after sowing of two *fpa* lines (SAIL 849-F10 and SAIL 720-B10) and Col-0.

**Figure 8 genes-10-00091-f008:**
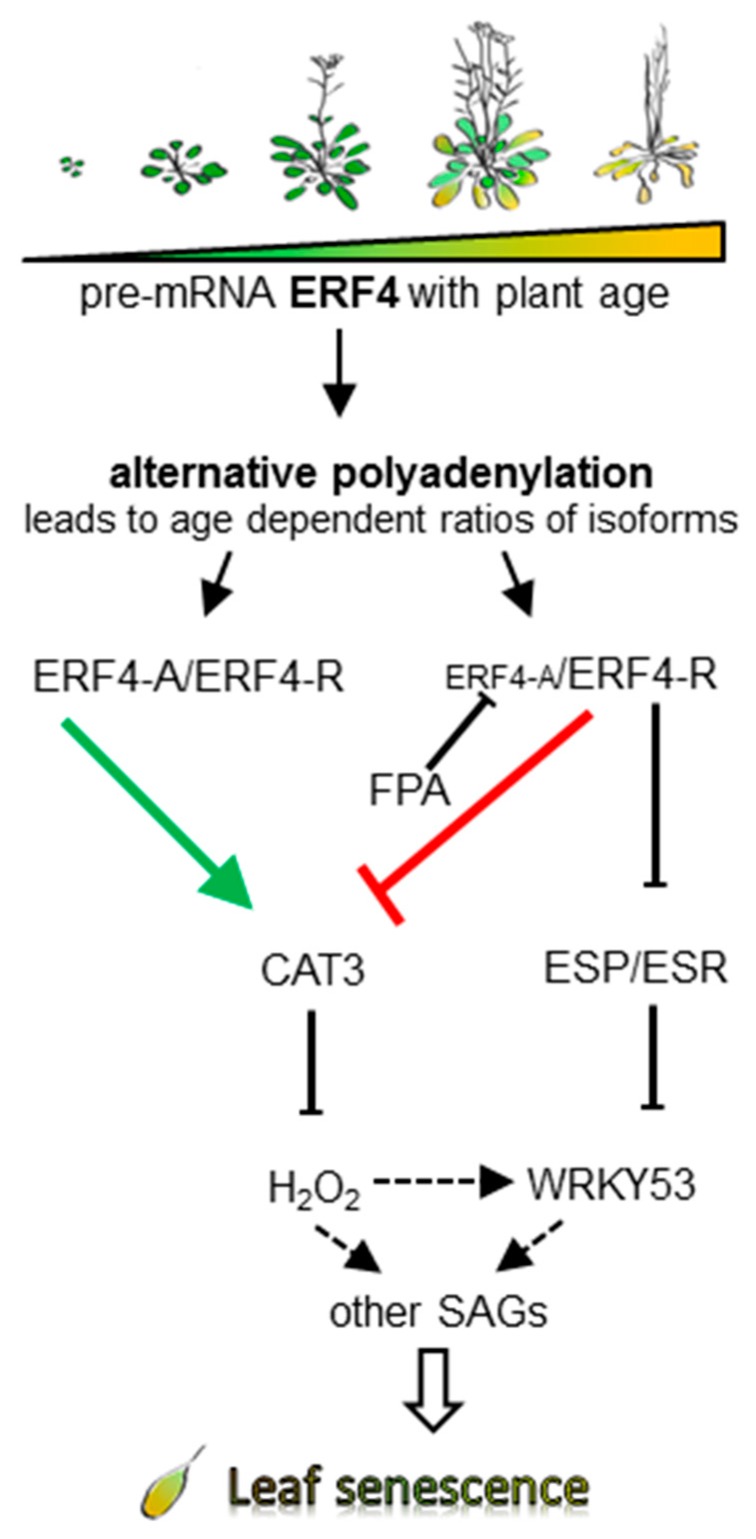
Model of the impact of the ERF4 isoforms in early and late senescence. Alternative polyadenylation, influenced by the RNA-binding protein FPA, leads to different ratios between the two isoforms in early and late senescence. FPA, which increases with age, leads to a reduced amount of ERF4-A in older plants, and consequently reduces the activation potential of ERF4-A on *CAT3*. Increasing H_2_O_2_ levels trigger expression of *WRKY53* and other senescence-associated genes (SAGs) and thereby leaf senescence. ERF4-R acts as a negative regulator of *CAT3* and *ESP/ESR* and thereby indirectly as a positive regulator of *WRKY53* expression and activity.

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
