# Peer review of "Impact of Alternatively Polyadenylated Isoforms of ETHYLENE RESPONSE FACTOR4 with Activator and Repressor Function on Senescence in *Arabidopsis thaliana* L."

_genes, 2019, doi:10.3390/genes10020091_

Round 1

Reviewer 1 Report

The manuscript “Impact of Alternatively Polyadenylated Isoforms of ETHYLENE RESPONSE FACTOR4 with Activator and Repressor Function on Senescence in Arabidopsis thaliana L” describes the analysis of two different isoforms of ERF4 in plant senescence. The results showed that one of the isoforms ERF4-R, acted as a repressor while the other form, ERF4-A, acted as activator of senescence. The authors also claimed that CAT3 was a direct target gene of ERF4 by showing that both isoforms directly bind to the CAT3 promoter but have antagonistic effects on gene expression. The manuscript is of significance in revealing the role of alternative polyadenylation in regulating plant senescence. I would suggest the following for the authors to consider in further improvement of this article:

The exist and ratio mRNA from of both isoforms are critical for the conclusions of ERF-A and ERF-R’s function. I would suggest the authors to include results of isoform specific RT-PCR in all related backgrounds including erf4, 35S:ERF4-A, 35S:ERF4-R, and the complementation lines.

For the protein stability analysis showed in Figure 2D and Figure S3B, controls showing equal loading are lacking.

For the claim “ERF4-R, acts as repressor due to the presence of an ERF-associated amphiphilic repression (EAR) motif …”, the authors did not show the function of the EAR motif specifically. This needs to be rephrased.

The two isoforms function antagonistically but both were able to complement the delayed senescence of erf4. The authors should discuss more on this discrepancy in the discussion part.

In the title “ETHYLENE RESPONSE FACTOR4” refers to the protein, so should not be in the italicized form.

Author Response

Resubmission of manuscript genes-420745

Lena Riester, Siliya Koester-Hofmann, Jasmin Doll, Kenneth Berendzen, Ulrike Zentgraf*

Response to reviewers

January 22nd, 2018

We would like to thank the reviewers for their comments on our manuscript. We address all of the reviewers’ concerns below and changed the manuscript accordingly. We feel that the revised version benefited from the changes. Please find below our responses to the specific issues raised. Changes are indicated via the “track changes function” in the word document.

Response to Reviewer #1

1.     mRNA of ERF4 isoforms in different backgrounds

The exist and ratio mRNA from of both isoforms are critical for the conclusions of ERF-A and ERF-R’s function. I would suggest the authors to include results of isoform specific RT-PCR in all related backgrounds including erf4, 35S:ERF4-A, 35S:ERF4-R, and the complementation lines.

We added representative results of the ERF4-isoform specific sqPCR in erf4 knock-out and overexpression lines in Figure S3B and of the complementation lines in Figure S4.

2.     Loading controls for western blots

For the protein stability analysis showed in Figure 2D and Figure S3B, controls showing equal loading are lacking.

In Figure 2D and S3B we show results of the immunodetection of ERF4 isoforms. We now added the amido black staining of the upper part of the PVDF membranes as loading controls that show equal loading below the Western Blots.

3.     Rephrasing and clarification of statement

For the claim “ERF4-R, acts as repressor due to the presence of an ERF-associated amphiphilic repression (EAR) motif …”, the authors did not show the function of the EAR motif specifically. This needs to be rephrased.

There was a sentence in the abstract, which implied causality between the EAR motif and the repressor function of ERF4-R, which was shown by other researchers but not specifically by us in our experiments, which are the basis for this manuscript. Therefore, we rephrased it in order to be more precise.

4.     Discrepancy between antagonistic function of ERF4 isoforms and complementation of delayed senescence phenotype

The two isoforms function antagonistically but both were able to complement the delayed senescence of erf4. The authors should discuss more on this discrepancy in the discussion part.

We added a new paragraph in the discussion section on how this can be explained.

5.     Editing

In the title “ETHYLENE RESPONSE FACTOR4” refers to the protein, so should not be in the italicized form.

We corrected the wrong italicized form.

Reviewer 2 Report

The manuscript submitted by Reister et al. is an interesting work, the experimental design is sound and the conclusions are supported by the experiments. 

I would ask the authors to include, in figure S1, a picture of all the leaves of the rossette (1 to 12 or 16 according to Fig. S1C) of the different plants (Col0, erf4, and OE lines), not only leaves 1 to 10. In this way we could also observe the young leaves. 

In line 452: is dominant... or dominates... instead of is dominate

Author Response

Resubmission of manuscript genes-420745

Lena Riester, Siliya Koester-Hofmann, Jasmin Doll, Kenneth Berendzen, Ulrike Zentgraf*

Response to reviewers

January 22nd, 2018

We would like to thank the reviewers for their comments on our manuscript. We address all of the reviewers’ concerns below and changed the manuscript accordingly. We feel that the revised version benefited from the changes. Please find below our responses to the specific issues raised. Changes are indicated via the “track changes function” in the word document.

Response to Reviewer #2

1.     Picture of rosette leaves

I would ask the authors to include, in figure S1, a picture of all the leaves of the rossette (1 to 12 or 16 according to Fig. S1C) of the different plants (Col0, erf4, and OE lines), not only leaves 1 to 10. In this way we could also observe the young leaves. 

We included all leaves of the rosette in Figure S1G so that also the young leaves can be seen.

2.     Editing

In line 452: is dominant… or dominates… instead of is dominate

We corrected the wrong spelling.

Round 2

Reviewer 1 Report

The authors have made all the requested changes.